# The Power of Active Listening to Address Medication Non-Adherence During Care Transition: A Case Report of a Polypharmacy Patient with Type 2 Diabetes

**DOI:** 10.3390/pharmacy13030064

**Published:** 2025-04-30

**Authors:** Léa Solh Dost, Giacomo Gastaldi, Marie P. Schneider

**Affiliations:** 1School of Pharmaceutical Sciences, University of Geneva, 1205 Geneva, Switzerland; 2Institute of Pharmaceutical Sciences of Western Switzerland, University of Geneva, 1205 Geneva, Switzerland; 3Pharma24, Living Lab and Academic Community Pharmacy, 1205 Geneva, Switzerland; 4Division of Endocrinology, Diabetes, Hypertension and Nutrition, Department of Medicine, Geneva University Hospitals, 1205 Geneva, Switzerland

**Keywords:** medication adherence, interprofessional collaboration, patient partnership, continuity of patient care, communication, case report

## Abstract

This case report explores the experience of a newly diagnosed type 2 diabetes (T2DM) patient transitioning from passive non-adherence to active adherence over a two-month period following hospital discharge. During this two-month period, he participated in four exploratory, non-interventional research interviews investigating his medication adherence, beliefs, and self-management strategies. His feedback on these research interviews highlighted the role of key communication strategies—such as patient partnership, non-judgmental communication, and interprofessional active listening—in fostering self-reflection and behavioural change. While these techniques are well-documented, there remains an urgent need to translate them into routine practice by integrating behavioural science and interprofessional collaboration into healthcare pregraduate and postgraduate education in order to increase awareness and skills in clinical environments.

## 1. Introduction

Type 2 diabetes mellitus (T2DM) is a complex and prevalent condition, affecting over 462 million people worldwide [1]. Managing T2DM is particularly challenging due to its frequent association with multiple comorbidities, requiring patients to make daily decisions about their care, including medication and insulin management [2,3]. The American Diabetes Association (ADA) reported that diabetes self-management education and support (DSMES) should be provided at different times in the patient journey, including when transitions in life and care occur [4].

Transitioning from hospital to home is a high-risk phase for patients with complex chronic conditions, such as T2DM [5,6,7]. Patients with T2DM face a higher risk of readmission due to their associated comorbidities, polypharmacy, and the use of medications with a narrow therapeutic index, such as insulin and oral hypoglycaemic agents [7,8]. After discharge, these patients frequently face discontinuity of care, insufficient medication management, and difficulties in medication adherence, all of which increase the patient’s vulnerability [9,10,11,12]. As up to 40% of hospitalised patients have T2DM, healthcare professionals should pay special attention to ensuring effective care management and anticipate various medication-related issues upon discharge [13,14].

Medication adherence is “‘the process by which patients take their medications as prescribed” [15]. Adherence rates for oral hypoglycaemic agents among T2DM patients vary widely, ranging from 36% to 93% depending on the population studied [16]. Thus, a substantial number of patients are non-adherent to their medications, which is associated with disease deterioration, increased comorbidities, and higher healthcare costs [17]. This issue is particularly critical at the time of hospital discharge: within one week, 28% of patients had not initiated at least one of their newly prescribed medications [18], and by 30 days post-discharge, nearly 40% had either not initiated or had already discontinued at least one medication [19]. The World Health Organisation (WHO) lists five main categories of determinants that influence medication adherence: patient-related, condition-related, treatment-related, health system-related, and socioeconomic factors [20].

Effective clinician–patient communication can improve health outcomes but is sometimes lacking during the transition of care [21,22,23]. Motivational interview, a method to explore patients’ ambivalence and enhance motivation in an interpersonal relationship, is one intervention to support and improve medication adherence in patients with chronic medications [24,25,26,27]. Within this framework, active listening, also known as reflective listening, is a crucial tool that aims to accurately understand patients by deeply engaging in their conversation, being attentive to their words, looking out for verbal and non-verbal cues, encouraging them to express themselves and reformulating their words to ensure understanding and encourage personal exploration [24].

Through a concrete and rare example, this case report aims to illustrate how active listening can positively influence a patient’s behaviour change and promote medication adherence in a newly diagnosed T2DM patient discharged from the hospital. By focusing on this single case, the study offers in-depth insight into the complex and dynamic process through which the patient gradually transitioned from passive non-adherence to active adherence behaviour, highlighting the practical impact of communication strategies.

## 2. Methods

The CARE case report guideline (Appendix A) was adapted in the Methods section [28].

### 2.1. Patient Information

A 45-year-old patient was hospitalised in March 2021 for eight days following a diagnosis of inaugural T2DM, with a blood glucose level of 29.3 mmol/l at admission, HbA1c of 13%, and a BMI of 32 kg/m^2^. His past medical history included hypertension, dyslipidemia, and a non-ST-segment elevation myocardial infarction (NSTEMI), all diagnosed two years earlier. From a socioeconomic standpoint, the patient had completed high school, was unemployed, received social welfare support, and lived in a shared apartment. The patient was discharged with the following medications: clopidogrel (75 mg, once daily), acetylsalicylic acid (100 mg, once daily), valsartan (160 mg, once daily), metoprolol (25 mg, once daily), rosuvastatin (10 mg, once daily), metformin (1000 mg, once daily), insulin aspart (10 IU, three times daily), and a combination of insulin degludec and liraglutide (30 IU, once daily).

During hospitalisation, the patient reported a history of medication non-adherence, having discontinued all prescribed treatments several months prior. He identified several key reasons for this decision:Routine incompatibility: his prescribed medication regimen included morning and evening doses, which conflicted with his sleep pattern of 5 am to 12 pm.Fear of drug interactions and side effects: he expressed concerns about potential adverse effects and interactions between medications.Psychological resistance: he reported that taking medication made him feel “like an old man”, struggling with the impact of his treatment on his self-perception and identity.

### 2.2. Intervention

This patient participated in a qualitative longitudinal study that explored medication beliefs, concerns, adherence, and care journeys of patients with type 2 diabetes mellitus (T2DM) and multiple comorbidities following hospital discharge. The primary research articles provide detailed information on the study methodology [9,10].

Eligible patients in this study were recruited during their hospitalisation based on the following criteria: T2DM, the presence of at least two additional comorbidities, and self-management of medications after discharge. Each participant completed four semi-structured, non-interventional interviews (mean duration: 46 ± 4.5 min) over two months post-discharge, conducted on approximately days 3, 10, 30, and 60. Interviews took place either at the participant’s home, at the university, or by phone/videocall, depending on their preference. For the patient in this case report, interviews took place face-to-face. The interviews were conducted by a pharmacist and researcher trained in qualitative methods and basic motivational interviewing, who adopted a neutral, non-judgmental attitude, did not provide therapeutic guidance, and presented herself as a researcher (not as a healthcare professional). The interview guides were developed using the WHO’s five dimensions of adherence, the Information-Motivation-Behavioural Skills model, and Social Cognitive Theory [20,29,30]. The interviews explored patients’ beliefs about medications, concerns, perceived side effects, adherence behaviours, and challenges in medication management. Example questions included: “How are you currently managing your medications?”, “How do you feel about the prescribed medications?”, and “Can you describe a time when you found it challenging to take your medication or missed a dose?”. Audio-recordings were transcribed verbatim, anonymised, and thematically analysed using Braun and Clarke’s reflexive thematic analysis approach [31,32]. This study was approved by the Cantonal Research Ethics Commission of Geneva (CCER, Ref. 2020-01779), and written informed consent was obtained from all participants.

A month after the research interviews, the patient went to his referent diabetologist consultation, in which the patient spontaneously expressed the benefits of the research interviews.

### 2.3. Data Collection and Analysis of Patient’s Testimony

When the patient initiated the discussion about the research interviews, the diabetologist sought and obtained the patient’s consent to audio-record his testimony. The audio-recorded testimony was transcribed verbatim and analysed using inductive (data-driven) thematic analysis. Emerging themes related to factors impacting medication adherence and skills to address them are reported in Section 3. Patient (non)adherence determinants were classified according to the five determinant categories listed in the WHO framework [20].

## 3. Results

### 3.1. Determinants of Patient Non-Adherence

The patient expressed doubts about his T2DM diagnosis and the true need for medications. He balanced their benefits and risks and expressed difficulty with multiple daily intakes and how this triggered his medication non-adherence. The patient did not present with any prominent literacy-related difficulties, but a critical factor in building his adherence was his understanding of the therapeutic indications and mechanisms of action for his medications (valsartan and clopidogrel). He also discussed past negative experiences with healthcare professionals, which continued to impact his trust in the healthcare system while developing a new trustful relationship with his current diabetologist. Figure 1 (*Classification of the patient’s non-adherent determinants*) organises the factors influencing his non-adherence according to the five determinant categories listed in the World Health Organisation (WHO) framework [20].

### 3.2. The Importance of Dedicated Time to Discuss Medications

The research interviews helped him to reflect on his medication use, adherence, and beliefs. The patient was able to elaborate on his medications, their usefulness, and how his beliefs and past negative experiences affected his actual medication adherence. Here is the patient’s feedback:

*“Hearing me talk about my use of medications [during the research interviews] and how psychologically apprehensive I was about taking them, made me realise how they useful were”*—patient, recorded testimony.

The interviews also helped him recognise that he had misunderstood the purpose of a particular medication, which had contributed to his non-adherence. Specifically, he believed that “valsartan” (an angiotensin receptor blocker) was a platelet inhibitor, leading him to discontinue its use, thinking it was unnecessary. After gaining a clearer understanding of the medication’s actual role, he took steps to comprehend the associated issues and risks better:

*“I stopped the drug [valsartan] for two weeks […] When I talked with [the researcher] about not knowing the indication, […] then I went to the pharmacy to take my blood pressure, and I realised that […] it was dangerous not to take it”*—patient, recorded testimony.

By devoting time to discussing this medication, the patient became more aware of its importance, which led to a change in his medication adherence behaviour. Before these interviews, the patient explained he had never found an opportunity to openly discuss medication non-adherence, beliefs, and motivation with healthcare professionals without judgement.

### 3.3. The Importance of Communication Skills

The trusting environment during the research interview was fostered through a collaborative, patient-as-partner approach, tailoring the discussion to the patient’s pace and maintaining a non-judgmental communication style. This approach enabled the patient to feel actively involved in the decision-making process, leading him to describe his central role in the discussions:

*“But the solution came from me. [...] At no time did anyone come to me and say, “You have to do this”. It came from me”*—patient, recorded testimony.

The interviewer adopted an active listening approach, using open-ended questions and reflective listening to authentically explore the patient’s concerns, validate his experiences, and encourage change talk. As shown in Table 1 (*Communication techniques used during research interviews*), the researcher’s use of brief prompts and strategic silences allowed the patient to further reflect on his situation and articulate a commitment to addressing his challenges.

## 4. Follow-Up

Since the study, the patient has attended all subsequent visits. T2DM has been well controlled since 2021. HbA1c values have remained between 6.5% and 7.5%, apart from one episode of hyperglycaemic decompensation in 2023 linked to the use of prednisone for severe back pain. Blood pressure medication has been adapted on several occasions, and the patient is now willing to increase the number of pills and medications.

## 5. Discussion

This case report involved a middle-aged patient hospitalised for T2DM with pre-existing comorbidities and a history of medication non-adherence who was reluctant to take some of his medications at discharge. Over the course of four non-interventional interviews, the patient progressed from a state of passive non-adherence to active engagement with his medication regimen. His self-reported reflections and concrete actions, such as reinitiating treatment and taking actions to monitor his blood pressure, provide evidence of a behavioural shift. These early signs of engagement are particularly significant, as the initial weeks following hospital discharge are a critical window for establishing long-term medication-taking habits [33,34]. This change is further supported by his now-stabilised type 2 diabetes and increased frequency of follow-up visits.

The novelty of this case report lies in demonstrating that well-established communication techniques, particularly active listening and open-ended questions, can have a lasting impact on patient adherence in a non-interventional study [24,35,36]. Such techniques could have an even more significant impact when combined with interventions based on shared decision-making and patient empowerment [37]. While the improvements in adherence observed in this case study resulted from in-depth interviews, these methods should be incorporated and evaluated into shorter, repeated interactions, such as short 15 min motivational interviews [38] or brief medication refills at the pharmacy. Increasing the integration of these techniques into short, targeted steps within a well-defined professional approach can enhance patient engagement and improve adherence outcomes.

Creating a trusting environment through these communication techniques, combined with a collaborative, patient-as-partner approach, allows patients to feel genuinely heard and provides them with the space to explore and express their beliefs and concerns about their medications [39]. In this case study, the patient spontaneously recognised the value of having a safe and supportive space to explore and openly express his beliefs and concerns about his medications. Within this environment, the patient developed an understanding of his medications and beliefs, effectively addressing barriers to adherence. Studies involving healthcare professionals across disciplines, including physicians, pharmacists, and nurses, emphasise the essential role of empathy, open communication, shared decision-making, and trust in building this foundation [23,35,40]. Repeated interactions that incorporate these elements contribute to building a lasting rapport, ensuring continuity of care, and promoting patient education, ultimately enhancing adherence [41].

Given that non-adherence is a modifiable risk factor, addressing patients’ perspectives on their medications is regularly needed, and is especially crucial after hospital discharge [18,42]. Improving medication adherence requires a proactive, system-level approach involving both patients and healthcare professionals. Supporting patients in this context calls for coordinated, targeted interventions across the care continuum. Physicians, as prescribers, play a central role in initiating treatment, fostering trust, and providing clinical rationale. At the same time, community pharmacists, as dispensers, are uniquely positioned to reinforce this information, detect misunderstandings, and offer practical guidance at the point of care during their frequent interactions [43,44]. The New Medicine Service, originally developed in England, is an example of an effective pharmacy-based interprofessional intervention for supporting medication initiation. In this model, community pharmacists conduct two brief, semi-structured interviews with patients to enhance their understanding of the treatment, reinforce self-efficacy, and identify early barriers to medication adherence [34]. While awareness of the importance of adherence continues to grow and adherence monitoring is now integrated into several clinical guidelines (e.g., for hypertension and type 2 diabetes) [45,46], there is an urgent need to shift from awareness to actionable implementation. This requires equipping healthcare professionals with behavioural science competencies, establishing clear standards for addressing medication adherence, and fostering interprofessional collaboration as a core element of routine care [47,48,49].

### 5.1. Limitations

This case report focuses on a single patient, with data primarily derived from his testimony at a specific time point. Although subsequent clinical indicator (such as the stabilisation of T2DM and increased frequency of follow-up) suggest a sustained behavioural change, it is likely that additional factors also contributed to the observed improvements in self-efficacy and long-term medication adherence. Additionally, the case was conducted within the structured and reflective environment of a research framework, which may limit the generalisability of the findings to routine clinical settings. Nonetheless, the communication techniques employed are well-supported in the literature and are applicable into routine care. Further research is needed to assess their long-term effectiveness and to identify optimal strategies for their implementation and adaptation in real-world clinical practice. 

### 5.2. Recommendations for Practice

To promote medication adherence within a typical 15 min consultation, healthcare professionals can apply the following strategies:Use active listening at key moments (e.g., medication changes, life transitions): ask open-ended questions, such as “*How have you felt about taking your medications?*” and allow pauses for patient reflection. This approach helps to uncover adherence barriers quickly.Focus on shared decision-making around key adherence issues: prioritise and address one key adherence challenge per visit. For example, if patients forget their evening dose, collaboratively explore solutions like setting reminders or adjusting timing to fit their routine. This empowers patients and makes interventions more effective.Optimise professionals’ skills: encourage healthcare professionals to play a key role based on their expertise. If some expertise is shared by the interprofessional team (e.g., patient education), the role and responsibilities of each healthcare professional must be defined.

## 6. Conclusions

This case report provides a valuable example of the power of patient-centred communication and active listening in improving medication adherence and beliefs, especially during vulnerable transition periods like hospital discharge. These findings reinforce the need to move beyond traditional adherence strategies and incorporate behavioural and communication science into chronic disease management in routine practice to improve patient engagement and medication adherence.

## Figures and Tables

**Figure 1 pharmacy-13-00064-f001:**
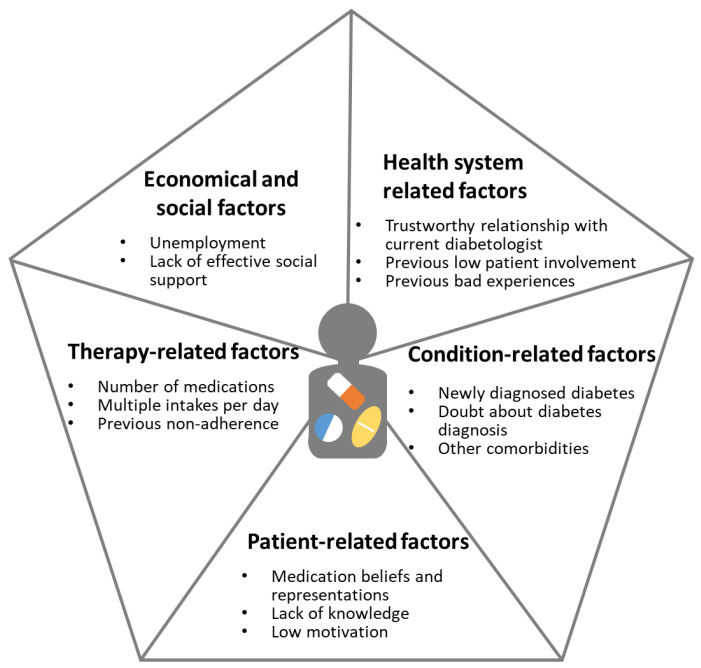
Classification of the patient’s non-adherent determinants based on the WHO classification [20].

**Table 1 pharmacy-13-00064-t001:** Communication techniques used during research interviews.

**Reflective listening**	**Patient:** I regularly forget to take this medication in the evening. **Interviewer:** Taking it in the evening can be tricky. **P:** It’s more like, from time to time I don’t take it [patient continues]
**Open-ended question**	**P:** Every time I see the four pills in my hands, immediately, I’m not feeling well…*[hesitation]* **I:** What do you feel? **P:** Well…there’s the fact of knowing I’m ill [patient continues]
**Eliciting change talk**	**P:** If this one is for blood pressure, maybe I should keep taking it… […] **I:** Now that you know the indication, it changes… **P:** Well…I think I will get my blood pressure monitored at the pharmacy…to see if my pressure has increased since I stopped the medication.

## Data Availability

The data presented in this study are available on request from the corresponding author due to privacy and confidentiality considerations regarding patient information.

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
