# Peer review of "The Power of Active Listening to Address Medication Non-Adherence During Care Transition: A Case Report of a Polypharmacy Patient with Type 2 Diabetes"

_pharmacy, 2025, doi:10.3390/pharmacy13030064_

Round 1
Reviewer 1 Report
Comments and Suggestions for Authors
Introduction and Rationale:
The introduction is good at setting up the subject matter. Explain briefly why single-case study offers value.
Methods
The qualitative methods are strong. Nevertheless, include a brief description of how interviews were structured and interviewer bias was controlled.
Results Presentation:
The results are nicely presented with helpful tables and graphs. Consider mentioning Figure 1 and Table 1 by name in your text to help make it clearer.
Discussion and Limitations:
The findings correspond with literature. It is clear to state that a single-case design's limitation must be mentioned and recommendations suggested for more studies.
Author Response
Dear reviewer,
Please find attached the point-by-point responses to your comment, under Reviewer 1.
Kindest regards,
The authors

Reviewer 2 Report
Comments and Suggestions for Authors
The article includes a case study of the adherents of a diabetic patient. Generally, I do not make any recommendations for the work structure. The authors analyzed the case of one patient in detail, indicating the causes of adherence problems and proposing a course of action. In the discussion, I propose to elaborate on possible ways of supporting the patient. In the authors' opinion, is the support of a doctor more important, or is the involvement of a pharmacist equally beneficial? The costs and effects of non-adherence are very high in Europe. Do the authors recognize the improvement role of pharmacists in these cases? Is it possible to provide additional recommendations for further action?
Author Response
Dear reviewer,
Please find attached the point-by-point responses to your comments, under Reviewer 2.
Kindest regards,
The authors

Reviewer 3 Report
Comments and Suggestions for Authors
Medication adherence, by the World Health Organization, represents "the degree to which the person’s behavior corresponds with the agreed recommendations from a health care provider." Adherence to prescribed medications signifies that the patient and physician collaborate to improve the patient’s health by integrating the physician’s medical opinion and the patient’s lifestyle, values, and preferences for care. Adherence will improve clinical outcomes for chronic disease management and reduce mortality from chronic conditions. Conversely, the consequences of nonadherence include higher rates of hospital admissions, suboptimal health outcomes, increased morbidity and mortality, and increased health care costs. In patients with polypharmacy, medication adherence is especially crucial, even more so after hospital discharge.
The manuscript presents a case report of a polypharmacy patient with type 2 diabetes where active listening positively influenced a patient’s behavior change and promoted medication adherence after discharge from the hospital.
The introduction briefly summarizes the medication adherence and issues related to adherence after hospital discharge. However, why this case is unique needs to be more clearly elaborated.
Check the abbreviations for type 2 diabetes mellitus (T2DM) and make them uniform throughout the manuscript (in some places, T2D was used).
The methods section is too scarce. Include a filled-out CARE checklist with the manuscript. Provide the exact age of the patient and the time period when he was hospitalized, which month. When were the comorbidities diagnosed? Were hypertension and the NSTEMI diagnosed two years ago?
Authors stated that "The primary research articles provide detailed information on the study's methodology [9, 10]." This is not enough. Please briefly and clearly explain the methodology. When was the interview done, exactly how many days post-hospital discharge? What are the interview characteristics? Who performed the interview? What were the interview guides? Provide data regarding ethics approval and informed consent in this section.
The result section is organized in three subsections: 3.1. Determinants of Patient Non-/Adherence, 3.2. The Importance of Dedicated Time to Discuss Medications, and 3.3. The Importance of Communication Skills. Why use this organization? Please explain; it is not clear from the methods.
"Figure 1. Classification of the patient’s non-adherent determinants according to the WHO classification (13)." – Check the reference; should this be reference 18 and not 13 in the brackets?
In the discussion, "These interviews had a lasting impact on the patient's adherence, even though he had been disengaged from his health for many years." – Can you say that this has a lasting impact when the follow-up was only a month after the research interviews? Provide and discuss the limitations in your approach to this case.
Author Response
Dear reviewer,
Please find attached the point-by-point responses to your comments, under Reviewer 3.
Kindest regards,
The authors

Reviewer 4 Report
Comments and Suggestions for Authors
The method of a case study seems appropriate for this manuscript. However, more information for the methods should be included. The intervention section is brief and leaves me with a few more questions. Did the interviews take place twice per week over two months? What was the communication method for the interviews (face-to-face, telephone, or virtual)?
In the results section, it is indicated (lines 117) that the patient "was highly literate", but the case presentation does not seem to coincide with the opinion of the authors and the behavior of the patient to initially stop taking his medication (my opinion).
I believe the communication techniques were adequately describe and elicited beneficial information that would lead one to think there would be a change in behavior on the patient's part. The aim indicates that the patient progressed from a passive state of non-adherence to "active adherence behaviour". Other than a response that "might" occur, I did not see the evidence for an actual change in behavior. Was there any information that provided evidence that adherence to the blood pressure medicine was going to take place actually took place (e. g. more comments from the patient)?
Author Response
Dear reviewer,
Please find attached the point-by-point responses to your comments, under Reviewer 4.
Kindest regards,
The authors

Round 2
Reviewer 3 Report
Comments and Suggestions for Authors
Authors had adequately responded to all comments and revised the manuscript accordingly. Manuscript has been improved significantly.
Reviewer 4 Report
Comments and Suggestions for Authors
Thank you for your revisions to the manuscript. I do not have any further concerns.